# OpenReview forum: "PROBABILISTIC RISK SCORING FRAMEWORKS FOR AUTOMATED CYBER INSURANCE UNDERWRITING: A BAYESIAN NETWORK APPROACH"
_mathai.club/MathAI/2026/Conference — 2026 Oral_

### Official Review · Reviewer_3Qgw · 2026-03-12
**Probabilistic Risk Scoring Frameworks for Automated Cyber Insurance Underwriting: A Bayesian Network Approach**

**Rating:** 6
**Confidence:** 4

**Review:**

Summary:
This paper proposes a probabilistic framework for automated cyber insurance underwriting based on Bayesian network inference. The approach models dependencies among observable security indicators, latent vulnerability states, and loss outcomes using a directed acyclic graph. The underwriting decision is formulated as a posterior inference over the predictive loss distribution, with closed-form solutions for conjugate exponential families and variational inference for the general case. The framework is further extended to dynamic Bayesian networks to capture the temporal evolution of cyber threats.

Strengths:
- The paper addresses an important real-world problem at the intersection of cybersecurity, finance, and probabilistic AI.
- The Bayesian network formulation provides an interpretable probabilistic structure, which is particularly valuable in regulated domains such as insurance.
- The proposed framework integrates posterior inference, variational approximation, and temporal modeling in a coherent probabilistic modeling pipeline.
- The empirical comparison suggests improved calibration performance compared with traditional actuarial models while maintaining the interpretability of risk factors.

Suggestions for improvement:
The paper could be strengthened by:
- providing additional details about the experimental setup and datasets;
- expanding the discussion of model assumptions and parameter estimation procedures;
- clarifying the scope of the theoretical guarantees presented for the variational EM procedure.

Final Recommendation:
POSTED / Poster-style acceptance with minor revision

Overall, the paper presents a clear and practically relevant probabilistic modeling framework for cyber risk scoring. With additional experimental and methodological detail, the work could contribute to ongoing discussions on interpretable probabilistic AI methods in financial risk modeling.

---

### Official Review · Reviewer_jmi8 · 2026-03-13
**There are problems in the "PROBABILISTIC RISK SCORING FRAMEWORKS FOR AUTOMATED CYBER INSURANCE UNDERWRITING: A BAYESIAN NETWORK APPROACH" paper**

**Rating:** 5
**Confidence:** 3

**Review:**

This paper is devoted to solution of such important task as automated cyber risk scoring. Bayesian network inference over directed acyclic graphs (DAGs) allows authors to solve this task.

This paper has the following disadvantages:
1) Too small size of paper did not allow authors to explain formulas in detail.
2) It is necessary to add comparison with related works in the paper.

---

### Decision · Program_Chairs · 2026-03-14

**Decision:**

Accept (Oral)

**Comment:**

Dear Author(s),

On behalf of the Program Committee of the International Conference on Mathematics of Artificial Intelligence (MathAI 2026), we are pleased to inform you that your paper has been accepted for an oral presentation at MathAI 2026.

Your paper was evaluated through a rigorous two-stage review process involving both automated screening and expert review by members of the Program Committee. The reviewers recognized the quality and contribution of your work.

Presentation details:

- Format: Oral presentation (15–20 minutes + 5 minutes Q&A)
- Mode: You may present either in person (offline) at the conference venue in Sirius, Russia, or remotely via Zoom. Please indicate your preferred mode when confirming your participation.
- Conference dates: Marh 30 - April 3, 2026
- Website: https://mathai.club

Next steps:

1. Please confirm your participation and presentation mode by replying to this email mathai.club@yandex.ru no later than March 15, 2026 18:00 Moscow time.
2. If you plan to attend in person, the organizing committee will provide accommodation details separately.
3. Please prepare your final camera-ready manuscript according to the formatting guidelines available at https://mathai.club and upload it to OpenReview by March 15, 2026 18:00 Moscow time.

Should you have any questions regarding the program, logistics, or your presentation slot, please do not hesitate to contact us.

We look forward to your contribution to MathAI 2026.

With kind regards,

MathAI 2026 Program Committee
International Conference on Mathematics of Artificial Intelligence
https://mathai.club
OpenReview: https://openreview.net/group?id=mathai.club/MathAI/2026/Conference
Telegram: https://t.me/MathAI_club
Email: mathai.club@yandex.ru